# Diagnostic Accuracy of Procalcitonin upon Emergency Department Admission during SARS-CoV-2 Pandemic

**DOI:** 10.3390/antibiotics11091141

**Published:** 2022-08-23

**Authors:** Stefano Malinverni, Silvia Lazzaroni, Maïa Nuňez, Thierry Preseau, Frédéric Cotton, Delphine Martiny, Fatima Bouazza, Vincent Collot, Deborah Konopnicki, Stéphane Alard, Magali Bartiaux

**Affiliations:** 1Emergency Department, Centre Hospitalier Universitaire Saint-Pierre, Université Libre de Bruxelles, Rue Haute 322, 1000 Brussels, Belgium; 2Centre Hospitalier Universitaire Brugmann, Place A.Van Gehuchten 4, Université Libre de Bruxelles, 1020 Brussels, Belgium; 3Clinical Chemistry, Laboratoire Hospitalier Universitaire de Bruxelles-Universitair Laboratorium Brussel (LHUB-ULB), Université Libre de Bruxelles, Rue Haute 322, 1000 Brussels, Belgium; 4Department of Microbiology, Laboratoire Hospitalier Universitaire de Bruxelles-Universitair Laboratorium Brussel (LHUB-ULB), Rue Haute 322, 1000 Brussels, Belgium; 5Infectious Diseases Department, Centre Hospitalier Universitaire Saint-Pierre, Université Libre de Bruxelles, Rue Haute 322, 1000 Brussels, Belgium; 6Department of Radiology, Centre Hospitalier Universitaire Saint-Pierre, Université Libre de Bruxelles, Rue Haute 322, 1000 Brussels, Belgium

**Keywords:** COVID-19, pandemics, procalcitonin, SARS virus, community-acquired infections, emergency service, hospital, pneumonia, viral, community-acquired pneumonia

## Abstract

**Highlights:**

**Abstract:**

Introduction: Procalcitonin is a marker for bacterial diseases and has been used to guide antibiotic prescription. Procalcitonin accuracy, measured at admission, in patients with community-acquired pneumonia (CAP), is unknown in the current severe acute respiratory syndrome coronavirus 2 (SARS-CoV-2) pandemic. Objectives: To evaluate the diagnostic accuracy of procalcitonin to assess the need for antibiotic treatment in patients with CAP presenting to the emergency department during the SARS-CoV-2 pandemic. Methods: We performed a real-world diagnostic retrospective accuracy study of procalcitonin in patients admitted to the emergency department. Measures of diagnostic accuracy were calculated based on procalcitonin results compared to the reference standard of combined microbiological and radiological analysis. Sensitivity, specificity, positive and negative predictive values, and area under (AUC) the receiver-operating characteristic (ROC) curve were calculated in two analyses: first assessing procalcitonin ability to differentiate microbiologically proven bacteria from viral CAP and then clinically diagnosed bacterial CAP from viral CAP. Results: When using a procalcitonin threshold of 0.5 ng/mL to identify bacterial etiology within patients with CAP, we observed sensitivity and specificity of 50% and 64.1%, and 43% and 82.6%, respectively, in the two analyses. The positive and negative predictive values of a procalcitonin threshold of 0.5 ng/mL to identify patients for whom antibiotics should be advised were 46.4% and 79.7%, and 48.9% and 79% in the two analyses, respectively. The AUC for the two analyses was 0.60 (95% confidence interval [CI] 0.52–0.68) and 0.62 (95% CI, 0.55–0.69). Conclusions: Procalcitonin measured upon admission during the SARS-CoV-2 pandemic should not guide antibiotic treatment in patients with CAP.

## 1. Introduction

Severe acute respiratory syndrome coronavirus 2 (SARS-CoV-2) infection is associated with high rates of emergency attendance, hospitalization, and intensive care unit admission. Most cases present with mild symptoms, and a small proportion evolve to more severe presentations, such as oxygen-requiring pneumonia, acute respiratory distress syndrome, or fatal issues [1]. The most common symptoms are fever, fatigue, and dry cough. Less common symptoms include sputum production, anorexia, sore throat, chest pain, and nausea [1,2]. These symptoms are aspecific and are frequently observed in pneumonia caused by other viruses and bacteria. Differentiating between viral and bacterial pneumonia or bacterial coinfection of viral pneumonia is challenging because of the overlap in presentation between these entities [3,4]. The choice of administering broad-spectrum antibiotics to these patients is difficult, and overprescription of antibiotics has been reported in hospitalized patients with SARS-CoV-2 [5]. While a delay in antibiotic treatment of bacterial community-acquired pneumonia (CAP) is associated with increased mortality [6], systematic broad-spectrum antibiotic treatment of suspected bacterial CAP is associated with complications, side effects, and mortality [7].

Procalcitonin (PCT) is a prohormone produced by the thyroid gland. In response to bacterial infection, interleukin-6 (IL-6), tumor necrosis factor-α (TNF-α), and interleukin-1β (IL-β) induce PCT synthesis in extrathyroidal tissue [8] with a peak at 6 h from the onset of infection and a half-life of 24 h [9]. In most viral infections, increased interferon gamma production inhibits PCT synthesis, leading to bacterial specificity of PCT.

Studies have suggested that PCT is a useful serum biomarker that supports clinical decisions regarding antibiotic treatment in patients with CAP. Higher serum PCT levels are associated with a higher probability of bacterial disease [10]. Some clinical trials have reported that PCT can guide clinicians in decisions on empiric antibiotic coverage without incurring higher rates of adverse outcomes [11,12], while others have reported no differences in antibiotic use among patients with suspected lower respiratory infection when PCT values were integrated in the treatment decision [4].

Within the context of the ongoing pandemic, PCT has been associated with the severity and mortality of SARS-CoV-2 infections [13,14,15]. Meanwhile, evidence on the role of PCT in guiding antibiotic prescriptions remains insufficient [16,17].

This two-center case–control study aimed to evaluate the role of PCT in differentiating CAP with an indication for antibiotic treatment from other entities associated with a new radiological infiltrate and lower respiratory tract infection (LRTI) signs and symptoms.

## 2. Material and Methods

### 2.1. Design

This retrospective two-center case–control observational study assessed the diagnostic accuracy of PCT for antibiotic prescription guidance in patients with CAP admitted to the Emergency Department of Saint-Pierre and Brugmann University hospitals between 1 March 2020 and 31 October 2020. The ethics committee of each hospital (OM 007 and OM 026) approved the study protocol and waived the need for signed informed consent due to the retrospective design of the study (CE20-12-11 and CE 2022/132).

### 2.2. Setting and Participants

A sample size of 225 was calculated as sufficient to detect specificity and sensitivity of 0.8 with a two-sided type I error of 0.05 and a power of 80%, assuming a prevalence of 20% and H_0_ of 0.6 for both sensitivity and specificity. For eligibility, we assessed a convenience sample of consecutive patient consultations and expected to achieve the previously calculated sample size within the study timeframe. Both centers routinely measure PCT levels of patients with suspected CAP upon admission. Patients with CAP having a serum PCT measurement performed within 24 h from ED admission and at least one viral and one bacterial investigation (a pair of hemocultures, sputum, or bronchoalveolar lavage) performed within 48 h from admission were included in this study. Minors, pregnant women, patients already on antibiotic treatment at the time of ED admission, and patients with an extrapulmonary site of infection diagnosed during their initial evaluation were excluded.

All enrolled patients had signs of acute infection (temperature of >38 °C, chills, altered mental status, and leukocyte count of >10,000/µL or <4000/µL). Moreover, all patients had at least one symptom of acute respiratory illness (cough, dyspnea, sputum production, tachypnea, pleuritic chest pain, ambient air oxygen saturation (SatO_2_) of <94%, or a loss of ≥4 SatO_2_ points following a 1 min walking test) or at least one finding during auscultation (crackles and rales). All included patients had a new infiltrate on radiological imaging performed within 48 h from admission. Missing data were treated as missing in the analysis, and no imputation was performed.

### 2.3. Outcome Measure and Analysis

PCT concentrations were measured by technicians blinded to the clinical information using Lumipulse G B•R•A•H•M•S PCT immunoreaction cartridges on a Lumipulse G600II instrument (Fujirebio, Ghent, Belgium).

Bacteriological analysis included culture from a respiratory tract specimen (sputum or bronchoalveolar lavage) or a pair of blood cultures. Diagnosis of SARS-CoV-2 infection was based on the COVID-19 Ag Respi-Strip^®^ (Coris Bioconcept, Gembloux, Belgium), followed by qRT-PCR in case of a negative result (RealStar^®^ SARS-CoV-2 RT PCR Kit, Altona Diagnostics), both performed on a nasopharyngeal swab [18]. Additional microbiological tests were performed according to clinical presentation. Additional bacterial testing included the *Legionella pneumophila* urinary antigen test and serological tests for *Chlamydophila pneumoniae* and *Mycoplasma pneumonia.* Additional viral testing included immunochromatographic techniques for influenza and adenovirus, direct fluorescence antibody tests for parainfluenza viruses, and inoculation of three cell cultures. Upon request, for 13 patients, respiratory tract specimens were analyzed using a multiplex PCR system to detect an additional fourteen viral and three bacterial targets (Biofire™ Filmarray™, bioMérieux, Marcy l’Etoile, France).

Chest radiographs were classified as showing radiographic evidence of pneumonia whenever a new infiltrate was mentioned in the X-ray report. Thoracic CT scans were considered to have radiographic evidence of pneumonia when the summary report mentioned probable bacterial or viral pneumonia. Radiological assessment was performed by a radiologist blinded to the PCT results. During the study period, thoracic low-dose CT scans for suspected CAP were additionally classified through a simplified classification as having typical features of bacterial CAP, typical features of SARS-CoV-2 CAP, intermediate features of both viral and bacterial CAP, or features evoking a diagnosis different from CAP.

CAP was defined as a new infiltrate on chest radiological study in a patient presenting with LRTI signs and symptoms [19]. Patients were classified according to laboratory test results and interpretation of radiological images. CAP for which antibiotic treatment was recommended was defined as cases with a microbiological analysis positive for pathogenic bacteria or with a chest CT lung infiltrate typical of bacterial pneumonia. CAP for which antibiotic treatment was recommended included bacterial CAP and viral CAP cases with a documented bacterial coinfection. CAP cases in which antibiotics were discouraged were defined as cases with a typical viral pneumonia infiltrate on radiological studies, with a microbiological analysis positive for a pathogenic virus and a lack of any microbiological analysis positive for a bacterial pathogen or imaging test suggesting a possible bacterial coinfection.

Sixty-eight (19%) patients could not be classified using this method. These cases were classified, in addition to the previously categorized patients, by two independent specialists blinded to the PCT results in a secondary analysis. These two specialists classified ambiguous cases according to clinical, microbiological, and radiographic results, antibiotic administration, and clinical evolution as CAP cases for which antibiotic treatment was recommended. In cases of disagreement, a third independent specialist provided a definitive classification.

Therefore, we retrospectively studied the accuracy of PCT in identifying CAP cases in which antibiotic therapy was recommended in two separate analyses, using the two aforementioned definitions for recommended or discouraged antibiotic treatment.

## 3. Analysis

### 3.1. Procalcitonin among Groups

We compared the PCT distribution within the two aforementioned analyses using the Wilcoxon rank-sum test. PCT cutoff values described in the literature as thresholds for identifying bacterial infections and guiding antibiotic therapy [11,12] were used to categorize CAP according to PCT values in four strata: <0.1 ng/mL, 0.1–0.249 ng/mL, 0.25–0.499 ng/mL, and >0.5 ng/mL.

### 3.2. Accuracy of Procalcitonin for Identifying Antibiotic-Requiring CAP

We calculated a nonparametric receiver-operating characteristic (ROC) curve for the two aforementioned dichotomous analyses to study the diagnostic accuracy of PCT in identifying antibiotic-requiring CAP. Sensitivity, specificity, negative predictive values, and positive predictive values were calculated using PCT cutoff values of 0.1 ng/mL, 0.25 ng/mL and 0.5 ng/mL. All statistical analyses were performed using Stata software version 16 (StataCorp, College Station, TX, USA).

## 4. Results

During the study period, 476 patients presented with complaints related to CAP. After application of inclusion and exclusion criteria, 359 (75.4%) patients were included in the current analysis (Figure 1).

Patient characteristics at inclusion are illustrated in Table 1.

Table 2 illustrates the etiological pathogens identified.

In total, 77 cases (26.5%) were classified as having CAP with an indication for antibiotic treatment in the primary analysis and 100 (27.9%) in the secondary analysis. All 214 (100%) patients with microbiologically documented viral CAP had SARS-CoV-2-related pneumonia.

PCT concentrations were higher in the CAP group where antibiotics were recommended (0.22 ng/mL; IQR, 0.11–2.22 ng/mL) than the CAP group for which antibiotic were discouraged (0.19 ng/mL; IQR, 0.1–0.39 ng/mL; *p* = 0.01) (Appendix A).

Antibiotic-requiring CAP was more prevalent in higher PCT strata. The prevalence of antibiotic-requiring CAP was 24% among patients with PCT <0.1 ng/mL and increased to 46% among patients with PCT ≥ 0.5 ng/mL (Appendix A). Results were ambiguous in the intermediate strata.

### 4.1. Accuracy of PCT for Identifying CAP Cases in Which Antibiotic Therapy Was Recommended

#### 4.1.1. Nested Cohort without Cases Classified According to Specialist Opinion

PCT performed poorly in identifying cases for which antibiotics were recommended (area under the curve [AUC], 0.60; 95% confidence interval [CI], 0.52–0.68) in the nested cohort, excluding patients classified according to specialist opinion (Figure 2).

A PCT threshold of ≥0.25 ng/mL to identify CAP for which antibiotics were recommended resulted in sensitivity of 48.1% (95% CI, 36.5–59.7%) and specificity of 61.7% (95% CI, 54.8–68.2%) (Table 3).

A threshold of ≥0.5 ng/mL to identify CAP in which antibiotics were recommended resulted in sensitivity of 41.6% (95% CI: 30.4–53.4%) and specificity of 82.7% (95% CI: 77.0–87.5%).

#### 4.1.2. Complete Cohort including Patients Classified According to Specialist Opinion

PCT performed poorly in identifying CAP cases for whom antibiotics were recommended [AUC: 0.62 (95% CI: 0.55–0.69)] in the entire cohort (Figure 3).

A PCT threshold of ≥0.25 ng/mL to identify CAP for which treatment with antibiotics was indicated resulted in a sensitivity of 50.0% (95% CI, 39.8–60.2%) and a specificity of 64.1% (95% CI, 57.9–69.9%) (Table 2). A PCT threshold of ≥0.5 ng/mL to identify CAP for which treatment with antibiotic was indicated resulted in sensitivity of 43% (95% CI, 33.1–53.3%) and specificity of 82.6% (95% CI, 77.5–87.0%).

We performed a sensitivity analysis nested on patients with SARS-CoV-2 to study the accuracy of procalcitonin in identifying a bacterial coinfection within CAP patients with SARS-CoV-2. Results were consistent with previous analysis. A threshold of ≥0.5 ng/mL to identify bacterial coinfections within SARS-CoV-2 CAP resulted in sensitivity of 40% (95% CI: 16.3–67.7%) and specificity of 80.4% (95% CI: 74.9–85.1%). PCT performed poorly in identifying bacterial coinfection (area under the curve [AUC], 0.60; 95% confidence interval [CI], 0.52–0.68) within SARS-CoV-2 CAP patients [AUC: 0.59 (95% CI: 0.44–0.76)].

## 5. Discussion

In this multicenter retrospective study of 359 adults with CAP admitted to the ED during the SARS-CoV-2 pandemic, including 298 with microbiologically documented pathogens, no PCT threshold identified CAP for which antibiotic treatment was recommended.

Trials and meta-analyses, partially performed in EDs, suggested that PCT could tailor antibiotic prescription in CAP without increasing adverse outcomes [11,12,20]. Guidelines based on previous results of trials provided graded recommendations based on four tiers of PCT levels and discouraged the use of antibiotics for patients with PCT values ≤0.1 ng/mL while strongly recommending antibiotics in patients with PCT values ≥0.5 ng/mL. Other studies have suggested thresholds of 0.2 ng/mL to differentiate CAP from bronchitis [21]. Studies performed during the H1N1 pandemic indicated that PCT could help distinguish patients with a bacterial etiology from those with viral pneumonia [22,23]. However, a recent trial cast doubt on the ability of PCT to reduce antibiotic exposure in ED-diagnosed CAP [4]. The results from our study, carried out during the SARS-CoV-2 pandemic, further challenged PCT-based recommendations to guide antibiotic administration in CAP. In this cohort, withholding antibiotic treatment in patients with PCT levels ≤0.1 ng/mL would have resulted in undertreating 17 (24.6%) cases of all CAP with a microbiological indication for antibiotic treatment. Moreover, the routine administration of antibiotics in our cohort in patients with PCT ≥0.5 ng/mL would have resulted in the inappropriate overtreatment of 37 (53.6%) of 69 patients (Appendix A). These high rates of over- and undertreatment, as well as the low AUC observed for PCT as a tool to identify antibiotic-requiring CAP from viral CAP, do not seem to support the role of PCT in guiding antibiotic prescription in CAP within the context of the SARS-CoV-2 pandemic.

Our results are in line with previous studies [4,10,24] and with recent American guidelines for the management of CAP that recommend against routine PCT measurements to determine the need for initial antibacterial therapy [25,26]. The first ROC analysis, restricted to patients for whom a causative microbiological pathogen was documented, reported an AUC of 0.60 without any clear cutoff point that could identify patients for whom antibiotic therapy should be recommended. PCT performed poorly in identifying cases for which antibiotics were recommended, suggesting that PCT alone is not sufficient in CAP to guide antibiotic prescription in the ED. While being methodologically sound to restrict the analysis of patients with a proven microbiological etiology or a typical focal, bacterial condensation on CT scans, results from this population are not directly generalizable to clinical practice, as pathogens may be detected in less than 40% of CAP cases. Moreover, thoracic CT is not routinely performed for the diagnosis of pneumonia [27].

The secondary analysis, including both CAP with a straightforward etiologic diagnosis and pneumonia classified according to an independent blinded review, showed a similar AUC of 0.62 for the ability of PCT to identify CAP with an indication for antibiotic treatment. This analysis yielded similar results in a more pragmatic clinical scenario where the clinician has to decide whether to initiate antibiotic treatment for CAP, irrespective of whether a pathogen will be documented from the microbiological analyses.

PCT is higher in SARS-CoV-2 CAP than in CAP associated with other viral etiologies [28]. In addition, PCT had extremely low (<10%) positive predictive values for bacterial pneumonia in a multicenter study investigating empiric antibacterial therapy for suspected SARS-CoV-2 CAP [16]. The lower discriminatory performance of PCT compared to that previously reported in the literature might be explained by the hyperinflammatory status and cytokine storm caused by SARS-CoV-2, resulting in higher PCT concentrations than in other viral CAP [14], thereby lowering the discriminatory power of PCT for bacterial infections. High levels of IL-1b and TNF-α, together with high IL-6, have been reported in SARS-CoV-2 patients [29] which might increase PCT plasma levels. Alternatively, PCT trajectories rather than absolute values have been suggested as possible markers of bacterial infection without solid evidence in favor of this practice [14,30]. This finding is relevant, as PCT-driven antibiotic prescriptions in the current pandemic context might drive antibiotic overconsumption and expose patients and hospitals to potential harmful effects. Moreover, the reported findings are in line with evidence suggesting that PCT might be a marker of severity in SARS-CoV-2 pneumonia, which would hinder its ability to be used as a diagnostic tool to withdraw or withhold antibiotic prescription [13].

## 6. Limitations

This study has both strengths and limitations. First, the retrospective design may have introduced a selection bias, as 24.6% of the patients with CAP were not included in the analysis according to the exclusion criteria (Figure 1). Second, the low proportion of CAP cases requiring antibiotic treatment might have reduced the diagnostic performance of procalcitonin. Third, while some studies recommend serial PCT measurements to guide antibiotic prescription in critically ill patients [30], we studied PCT at the time of ED admission. Fourth, the two-center design may limit the generalization, and fifth, the emergence of new SARS-CoV-2 variants that might elicit distinct procalcitonin responses. Finally, immunosuppression, which might alter cytokine expression in response to SARS-CoV-2 infection, could influence procalcitonin response to different CAP etiologies, and was not accounted for in this analysis.

The study’s main strength is its pragmatic design and the few exclusion criteria allowing us to interpret our results on PCT accuracy in identifying bacterial CAP in real-life ED conditions. The second strength is the high proportion of patients (83.0%) with a microbiologically documented etiology of CAP, which is higher than that found in previous PCT trials.

## 7. Conclusions

Procalcitonin measured at ED admission performed poorly as a guide for antibiotic prescription in CAP during the SARS-CoV-2 pandemic.

## Figures and Tables

**Figure 1 antibiotics-11-01141-f001:**
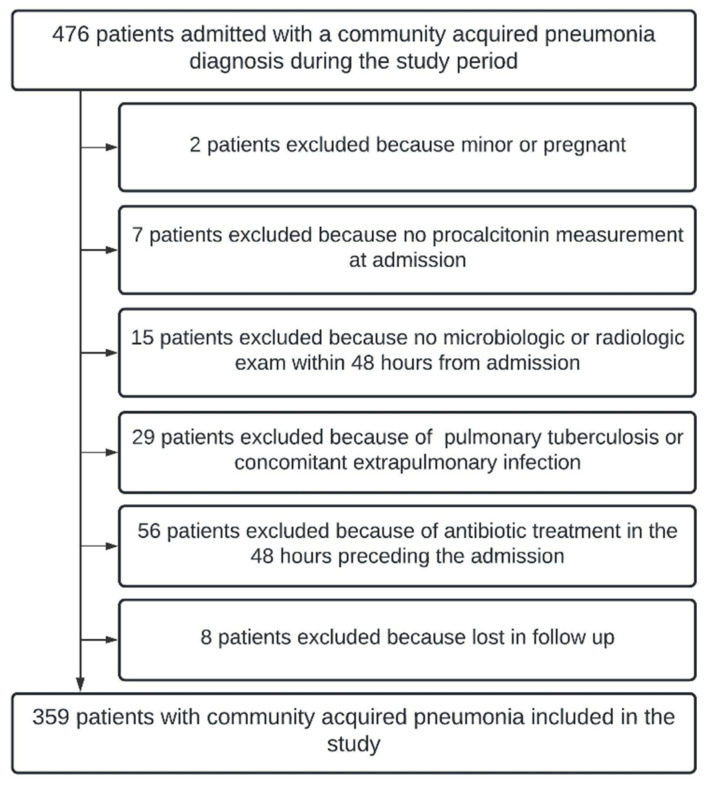
Study-inclusion flowchart.

**Figure 2 antibiotics-11-01141-f002:**
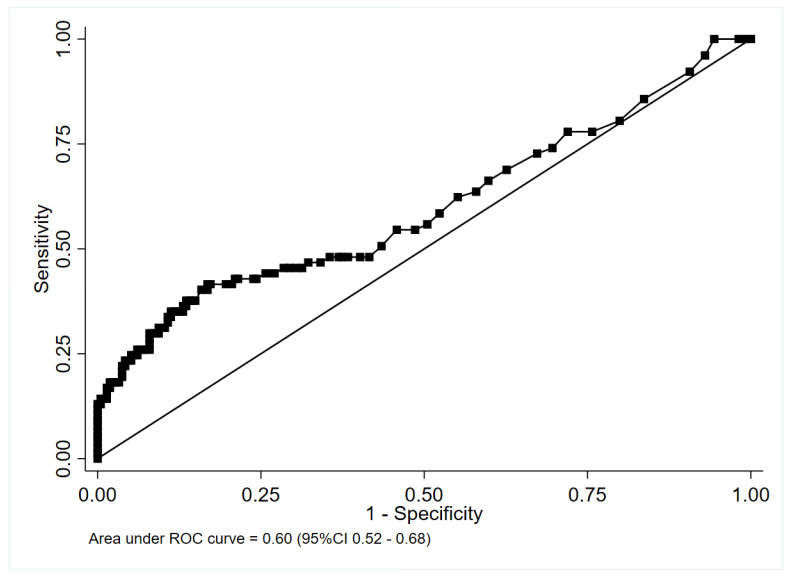
Receiver-operating characteristic curve of serum procalcitonin for the diagnosis of microbiologically proven bacterial pneumonia.

**Figure 3 antibiotics-11-01141-f003:**
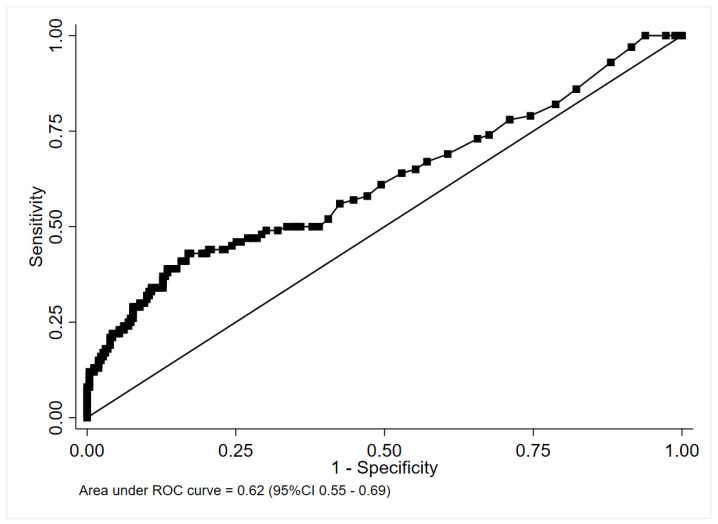
Receiver-operating characteristic curve of serum procalcitonin for the diagnosis of clinical bacterial pneumonia.

**Table 1 antibiotics-11-01141-t001:** Demographic and clinical characteristics at study inclusion.

*Number of Observations*	359
**Age, mean (IQR), y**	61 (49–75)
**Female, No. (%)**	112 (31.2)
**Days since first symptoms, median (IQR)**	7 (4–9)
**Coexisting conditions, No. (%)**	
**Hypertension**	148 (41.2)
**Diabetes**	105 (29.3)
**Chronic renal failure**	59 (16.6)
**COPD**	27 (7.6)
**Asthma**	24 (6.7)
**Chronic heart failure**	21 (5.9)
**Cerebrovascular disease**	17 (4.7)
**Nursing home resident, No. %**	40 (11.1)
**Pneumonia severity index, median (IQR)**	77 (58–105)
**Heart rate, median (IQR), beats/min**	99 (85–111)
**PaO_2_/FiO_2_, median (IQR)**	300 (245–340)
**Respiratory rate, median, (IQR), breaths/min**	24 (20–30)
**Systolic blood pressure, mean (IQR), mmHg**	130 (117–144)
**Blood urea nitrogen, median (IQR), mg/dL**	16 (11–22)
**Arterial pH, median (IQR)**	7.47 (7.45–7.50)
**Lactate, median (IQR), mEq/L**	1.1 (0.9–1.7)
**Procalcitonin, median (IQR) ng/mL**	0.19 (0.10–0.48)

Abbreviations: No., number; IQR, interquartile range; COPD, chronic obstructive pulmonary disease; PaO_2_/FiO_2_, arterial oxygen partial pressure to fractional inspired oxygen ratio.

**Table 2 antibiotics-11-01141-t002:** Etiological diagnosis using microbiological and molecular methods.

*Number of Observations*	359
**Identified pathogen, No. (%)**	297 (83)
** *SARS-CoV-2* **	244 (68)
** *Staphylococcus aureus* **	18 (5)
** *Hemophylus influenzae* **	9 (2.5)
** *Streptococcus pneumoniae* **	8 (2.2)
**Other**	28 (7.8)

Abbreviations: No., number.

**Table 3 antibiotics-11-01141-t003:** Cutoff levels, sensitivity and specificity for procalcitonin in detecting bacterial pneumonia.

Value	Cutoff Level ng/mL	Sensitivity, % (95% CI)	Specificity, % (95% CI)	Patients, No. (%)				PPV	NPV
				True Positive	False Negative	False Positive	True Negative	(95% CI)	(95% CI)
Procalcitonin [antibiotic-requiring vs. viral CAP] n = 291	>0.1	77.9 (67–86.6)	24.3 (18.7–30.6)	60 (20.6)	17 (5.9)	162 (55.7)	52 (17.9)	27.0 (21.3–33.4)	75.4 (63.5–84.9)
	>0.25	48.1 (36.5–59.7)	61.7 (54.8–68.2)	37 (12.7)	40 (13.8)	82 (28.2)	132 (45.4)	31.1 (22.9–40.2)	76.7 (69.7–82.8)
	≥0.5	41.6 (30.4–53.4)	82.7 (77.0–87.5)	32 (11.0)	45 (15.5)	37 (12.7)	177 (60.8)	46.4 (34.3–58.8)	79.7 (73.8–84.8)
Procalcitonin [confirmed and clinical bacterial vs. confirmed and clinical viral CAP] n = 359	>0.1	79.0 (69.7–86.5)	25.5 (20.3–31.2)	79 (22.0)	21 (5.9)	193 (53.8)	66 (18.3)	29.0 (23.7–34.8)	75.9 (65.5–84.4)
	>0.25	50.0 (39.8–60.2)	64.1 (57.9–69.9)	50 (13.9)	50 (13.9)	93 (25.9)	166 (46.2)	35.0 (27.2–43.4)	76.9 (70.6–82.3)
	≥0.5	43.0 (33.1–53.3)	82.6 (77.5–87.0)	43 (12.0)	57 (15.9)	45 (12.5)	214 (59.6)	48.9 (38.1–59.8)	79.0 (73.6–83.7)

Abbreviations: PPV, positive predictive value; NPV, negative predictive value.

## Data Availability

The complete dataset of the study will be available in the figsharer depository with the following DOI: https://doi.org/10.6084/m9.figshare.19283672.v1.

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
