# Peer review of "Diagnostic Accuracy of Procalcitonin upon Emergency Department Admission during SARS-CoV-2 Pandemic"

_antibiotics, 2022, doi:10.3390/antibiotics11091141_

Round 1
Reviewer 1 Report
Minor comments
1. Is there a project identification code or number of local Ethics committee?
2. Did the authors also evaluate the diagnostic accuracy of PCR in parallel with procalcitonin?
Author Response
Reviewer 1
Dear Reviewer, we thank you for the comments which indeed improved the manuscript.
Here under the answers to your comments
Minor comments
- Is there a project identification code or number of local Ethics committee?
We thank you for the comment, as requested we added the identification number of the local Ethics committee (OM 007 and OM 026) and the code of the approval (CE/20-12-11 and CE 2022/132)
- Did the authors also evaluate the diagnostic accuracy of PCR in parallel with procalcitonin?
We thank you for your comment. I must apologize but I am unsure about what you are referring to in terms of PCR.
In case you are referring to polymerase chain reaction and imply SARS-CoV-2 PCR: we used SARS-CoV-2 PCR as a gold standard for SARS-CoV-2 diagnosis. Because of this assumption we could not study the diagnostic accuracy of PCR.
In case you are referring to C-reactive protein: we studied CRP (PCR) and compared its performance to procalcitonin.
The AUC for CRP ability to identify CAP cases for whom antibiotics were recommended in the complete cohort including patients classified according to specialist opinion was extremely low: AUC 0.50 (95%CI 0.43 -0.57)
When comparing CRP to PCT in their ability to identify a bacterial infection in clinically and microbiologically classified CAP we obtained an AUC of 0.62 for PCT and 0.50 for CRP (p=0.0003).
Globally, our results showed that CRP was useless in identifying patients that could benefit from antibiotic treatment.
Nevertheless, given the well known low performance of C-reactive protein in identifying a bacterial etiology for CAP patients and given that the article was already long and complex we preferred not to add this further aspect to the manuscript. Nevertheless, if you think that this is required please tell us how do you thing that the results should be presented in the manuscript to facilitate an optimal understanding from the reader.

Reviewer 2 Report
Malinverni and co-authors submitted and article in which they report the results of a retrospective study performed in two Belgian centers, which aimed at determining the diagnostic value of procalcitonin in patients admitted to the emergency department during the SARS-CoV- 2 pandemic. The main results are that procalcitonin was of poor diagnostic value to discriminate between patients who should or should not have received antibiotics based on microbiological results and expert review for the cases which were difficult to categorize. The analysis seems to have been correcly performed, the article is well written and the message of the article is reasonably supported by the results. I would have the following comments.
Major comments :
- A sensitivity analysis would be required among the group of COVID-19 patients to study the value of procalcitonin for discriminating patients who have a bacterial coinfection or not. This has already been explored in the literature but should be explored in this cohort as well to reinforce the message in this specific setting.
- Was molecular diagnostic used for pneumonia etiological work-up ? If yes, please detail which panels.
- Please improve Table 1, particularly the microbiology subsection should be isolated and pathogens names italicized.
- Figures and Tables have no titles and legends
- Add one sentence in the limitations section to discuss the fact that SARS-CoV-2 variants, immunosuppression, may change the results, limiting the generalizability of your findings to current COVID-19 waves.
Author Response
Reviewer 2
Dear reviewer, we thank you for the comments that indeed improved the manuscript.
Here under the answers to your requests:
Major comments :
- A sensitivity analysis would be required among the group of COVID-19 patients to study the value of procalcitonin for discriminating patients who have a bacterial coinfection or not. This has already been explored in the literature but should be explored in this cohort as well to reinforce the message in this specific setting.
We thank you for the comment. Initially we considered that the two analysis presented were already reinforcing each other as a sort of sensitivity analysis. Nevertheless we thank you for the comment and we introduced a sensitivity analysis. As requested we studied the the value of procalcitonin for discriminating patients who have a bacterial coinfection or not within patients with COVID 19. We introduced the analysis as follows in the manuscript:
We performed a sensitivity analysis nested on patients with SARS-CoV-2 to study the accuracy of procalcitonin in identifying a bacterial coinfection within CAP patients with SARS-CoV-2. Results were similar to previous analysis. A threshold of ≥0.5 ng/ml to identify bacterial coinfections within SARS-CoV-2 CAP resulted in a sensitivity of 40% (95% CI:16.3% - 67.7%) and a specificity of 80.4% (95% CI:74.9% - 85.1%). PCT performed poorly in identifying bacterial coinfection (area under the curve [AUC], 0.60; 95% confidence interval [CI], 0.52–0.68) within SARS-CoV-2 CAP patients [AUC:0.59 (95% CI:0.44 – 0.76)].
- Was molecular diagnostic used for pneumonia etiological work-up ? If yes, please detail which panels.
We thank you for this question. Indeed a molecular diagnostic test was used for pneumonia etiological work-up in a minority of cases. We previously mentioned in the manuscript the following sentence: “Upon request, respiratory tract specimens were analyzed using a multiplex PCR system to detect an additional 14 viral and three bacterial targets.” In order to comply with your request we modified the sentence accordingly: “ Upon request, for 13 patients, respiratory tract specimens were analyzed using a multiplex PCR system to detect an additional fourteen viral and three bacterial targets (Biofire™ Filmarray™, bioMérieux) ”
- Please improve Table 1, particularly the microbiology subsection should be isolated and pathogens names italicized.
Thank you for the comment. We improved Table one by changing its title, adding two abbreviation that were not specified and by separating the microbiology subsection as suggested. Moreover pathogens names were italicized as suggested
- Figures and Tables have no titles and legends
We thank you for the comment. Titles were specified at the end of the manuscript. Nevertheless we tried to clarify as best as we could this issue. At the end of the manuscript we added Titles and Legends for each figure and table.
- Add one sentence in the limitations section to discuss the fact that SARS-CoV-2 variants, immunosuppression, may change the results, limiting the generalizability of your findings to current COVID-19 waves.
We thank you for this suggestion. As requested we added in the limitation section the two suggested limitations as follows:
Fifth, the emergence of new SARS-CoV-2 variants which might elicit distinct procalcitonin responses. Finally, immunosuppression, which might alter cytokine expression in response to SARS-CoV-2 infection, could influence procalcitonin response to different CAP etiologies and was not accounted for in this analysis.